# Portugal 2020: An Effective Policy Platform to Promote Sustainable Territorial Development?

**Eduardo Medeiros** 

DINÂMIA'CET - IUL, Instituto Universitário de Lisboa (ISCTE-IUL), 1649-026 Lisboa, Portugal; Eduardo.Medeiros@iscte-iul.pt

**Abstract:** This paper explores the implementation of European Union (EU) Cohesion Policy in Portugal during the 2014–2020 programming period (Portugal 2020) and its contribution to promoting sustainable territorial development. It starts with an anatomization of the dominant analytic dimensions of the concept of sustainable territorial development. It then examines the approved projects under Portugal 2020 and relates them to the selected five dimensions of territorial sustainable development, which include a circular economy, social environmental awareness, environmental conservation, a global governance sustainability focus, and a global spatial planning sustainability focus. The conclusions are that the Portugal 2020 goals of supporting sustainable territorial development have been, until late 2019, achieved beyond initial expectations in terms of relative allocated funding, and that the renewable energy and circular economy components have not been sufficiently explored, vis-à-vis the sustainable development needs of the country.

**Keywords:** sustainable territorial development; Portugal 2020; EU Cohesion Policy; territorial cohesion; circular economy; renewable energy

## 1. Introduction

Sustainable development has its roots in the underlying notion that natural resources are finite and, therefore, need to be explored in a sustainable way. As such, sustainable development policies can affect the survival of not only our species, but also many other species, and ultimately the biosphere of planet earth [1]. In this context, the strategy and the operationalization of public development policies are key to determining the relevance and potential effects of these policies (results and impacts–see reference [2]) for promoting sustainable territorial development processes [3]. Such policy concerns have propelled the inclusion of sustainable territorial development at the heart of the United Nations (UN) (i.e., 2030 Agenda for Sustainable Development) and the EU (i.e., Europe 2020) mainstream policy agendas [4].

The analysis focuses on the relevance of EU Cohesion Policy for promoting sustainable territorial development, due to its crucial importance for the territorial development process in the EU and, in particular, the selected case study: continental Portugal [5]. Crucially, EU Cohesion Policy aims to reduce regional development disparities across the EU. This is mainly done in terms of strengthening the economic and social dimensions of cohesion [6]. This policy instrument is formulated for all EU Member States, aiming to achieve similar results in structurally similar regions [7], that is, regions with similar socioeconomic development predicaments and challenges.

Demonstrably, in a country without elected regional government structures, like Portugal, (in the continental area), the regional development policies are mainly fueled by EU Cohesion Policy funding, in particular via the regional operational programs [8]. As of 30 September 2019, 69 B€ have been executed under Portugal 2020, 33% of which was assigned to the less developed continental region (North-Norte), whereas 24% was allocated to the Centre (Centro) region [9].

Despite its financial and policy relevance, EU Cohesion Policy has always been a controversial subject [6,10]. This controversy stems from evidence of its positive effects on territorial development in less developed regions [11–14], and also from noticeable doubts on its overall effectiveness [15–17]. Accounting for more than a third of the EU budget, EU Cohesion Policy has been a cornerstone of EU development policy [18]. It has also served as a concrete tool to implement ongoing EU development strategy visions, from place-based [19], multilevel governance [20], and smart specialization [21] narratives. Alongside these areas, it has also forged a connection with the European Spatial Development Perspective via the ESPON and INTERREG/European Territorial Cooperation programs [22], as well as with the Lisbon [23] and the Europe 2020 mainstream EU policy strategies [24].

Regarding the contribution of EU Cohesion Policy towards sustainable development, Pîrvu et al. [25] conclude that this policy has been used mostly as an economic growth tool, in particular in Central and Eastern European countries. These authors also highlight the mainstream policy focus of EU Cohesion Policy in supporting traditional socioeconomic development processes, which are linked to the United Nations (UN) sustainable development goals (SDGs). This analysis is, however, largely based on a statistical-based examination. Conversely, from a methodological standpoint, this paper is supported by a detailed project analysis, which fits each project into a novel proposed sustainable territorial development concept framework, thus adding a distinct flavor to available literature. Likewise, the proposed theoretical framework to link sustainable development and EU Cohesion Policy differs significantly from existing scientific rationales. This can be seen, for example, when reading a European Commission (EC) report, which relates the role of EU Cohesion Policy to four key sustainable development challenges: (i) climate change and clean energy; (ii) sustainable transport; (iii) conservation and management of natural resources; and (iv) sustainable consumption and production [26].

This broader multidimensional conceptual perspective is also engaged with recent literature on sustainable territorial development processes, which elevate, for instance, the importance of institutions and innovation, as driving forces for reaching a higher social welfare and for improving environmental quality [27]. Seeking to take stock of and consider this broader conceptual rationale in this paper, an effort is made to address the contributions of EU Cohesion Policy not only to environmental aspects associated with the concept of sustainable territorial development, but also to aspects associated with social awareness, a circular economy, global governance, and global spatial planning. This particular conceptual approach intends to stimulate beneficial debates on the topic of sustainable territorial development, making it relevant for practitioners, policy-makers, and scholars. In more detail, our research intends to address the tensions and the interplay between all the proposed territorial sustainable development dimensions. Building upon this integrative perspective, this paper can also provide a useful contribution to the development of the literature on sustainable territorial development, as well as to offer concrete and useful contributions to improve the effectiveness of EU Cohesion Policy in future programming periods, in particular in improving sustainable territorial development processes. The analysis will be driven by the following research questions: (i) In what ways has Portugal 2020 contributed to promoting a sustainable territorial development strategy? and (ii) Which analytic dimensions of sustainable territorial development were more positively affected by Portugal 2020, if any?

To better organize the analysis, the paper is structured as follows. The next section discusses the concept of sustainable territorial development and proposes a conceptual framework to be applied in the selected case-study: Portugal 2020. The third section is used to present the empirical analysis. Finally, the last section will detail Portugal 2020's direct contribution to a crucial component of environmental sustainability: the production of renewable energy. This is in line with the strategic relevance of this topic for Portugal because the country imports all its fossil fuels to propel its economy.

## 2. Sustainable Territorial Development: Analytic Dimensions

Sustainable development is, in our view, the most important policy concept of the twentieth century. Firstly defined in 1987, by the World Commission on Environment and Development, as the "development which meets the needs of the present without compromising the ability of future generations", sustainable development has been, for the most part, linked with three essential policy dimensions: (i) economic—to avoid extreme imbalances that damage industrial or agricultural production; (ii) environmental—to avoid overexploitation of renewable resource systems and to maintain a stable resource base; and (iii) social—adequate provision of social services and equality in distribution [28].

More recently, Sachs [29], reinforced this mainstream conceptual vision of sustainable development as being a part of an interaction of three complex systems: (i) the world economy: the need for widespread economic progress; (ii) the global society: the need to eliminate extreme poverty and strengthen the community; and (iii) the Earth's physical environment: the need to protect the environment from human-induced degradation. At the same time, however, Sachs added a fourth conceptual dimension to the debate of sustainable development: good governance. This was based on the realization that "governments must carry out many core functions to enable societies to prosper" [29] (p. 3).

Crucially, since 1987, the concept of sustainable development has been debated in so many different ways that its veracity has been largely undermined by a set of ambiguities [30,31]. It has also become commonplace to argue that the relative lack of conceptual precision of the notion of sustainable development, which is connoted with a certain analytical and political flexibility is, often times, argued to be an advantage in view of divergent viewpoints and interests to promote it [32]. The major critiques of sustainable development are a result of challenges from ongoing consumer patterns and neo-liberal interests [33].

Amongst many proposed definitions, sustainable development has been presented as a complex and multidimensional issue [34]. In many instances, it presents a major focus on intergenerational equity [34]. As in most cases, however, the bulk of the literature does not provide a clear distinction between the concepts of sustainable development and territorial development [35,36]. Indeed, the latter concept is also dependent on economic, social, environmental, and positive governance progress trends, as well as the implementation of sound spatial planning processes [37].

In essence, what distinguishes sustainable development and territorial development is the notion of sustainability (a goal or condition vis-à-vis the process of sustainable development—see [31] which needs to be associated with the notion that future generations have the right to enjoy a clean, socially and economically prosperous, and well-governed and planned world. For Harris and Goodwin [28], sustainability relates to a choice of goods and technologies oriented towards the requirements of ecosystem integrity and species diversity. According to Wall [32] (p. 395), based on Lew et al. [38], sustainability entails five crucial components: (i) an assumption that stability and balance is the norm; (ii) goals based on normative ideals of conservation, fairness and intergenerational sustainability; (iii) a research focus on the impacts of development and overuse of resources; (iv) methods of education for behavior change, recycling and greening, etc.; and (v) criticism that the concept is poorly defined and highly publicized.

For Despotovic et al. [39], sustainability encompasses the aforementioned three mainstream dimensions (social, economic, and environmental). More precisely, the authors invoke the need for promoting social sustainability (institutions, infrastructure, primary education, health, and the macroeconomic environment); efficiency enhancers (financial market development, training, higher education, technological readiness, labor market efficiency, and market size); and innovation and business. Seghezzo [40] presents a five-dimension sustainability triangle. Three dimensions (x, y, and z) relate to place. Permanence is added as the fourth dimension of time (t), whereas the last dimension (persons—p) adds an individual and human characteristic to the concept.

Regarding the social dimension of sustainable development, there is a general awareness that it is the one that has gained the least attention amongst the three mainstream pillars (economic, social, environmental; people, planet, and profit; environment, economy, and equity; or place, permanence, and persons) of the conceptual debate, since it is particularly difficult to realize and operationalize [41]. However, a cursory glance at the proposed components of the concept: (i) access to basic necessities (sanitation, drinking water, and healthcare); (ii) vulnerability to shocks (vulnerable employment, informal economy, and social safety net protection), and (iii) the social cohesion income gini index, social mobility, and youth unemployment [39], presents a case to mirror them with the social cohesion dimension of territorial development [42]. In a similar yet complementary way, Murthy [43] suggests four concrete social pillars for sustainable development: (i) equity; (ii) awareness for sustainability; (iii) participation; and (iv) social cohesion. The idea that sustainable development should be understood as ethically acceptable and socially just is presented as "a major normative regulation principle for contemporary society which includes a long-term ethical relationship of present generations with those of the future" [44] (p. 451).

As expected, the United Nations (UN) SDGs embrace a multidimensional perspective and constitute a good opportunity to reinvigorate the research on sustainable development. According to Filho et al. [45], these SDGs can be associated with six main thematic areas: dignity, people, planet, partnership, justice, and prosperity. These concerns place inequality at center stage of the SDGs [46]. In our view, however, they prompt us to confront their denomination which, in view of the proposed actions, are more appropriately connoted with the more encompassing goals of territorial development rather than sustainable development. Based on the above remarks, from our perspective, the following policy goals should be associated with the concept of sustainable territorial development (Figure 1):

1. A circular economy: aims to maintain the value of products, materials and resources for as long as possible by returning them to the product cycle at the end of their use, while minimizing the generation of waste [47];
2. Social environmental awareness: aims to foster an environmentally proactive and educated society [48];
3. Environmental conservation: aims to protect, restore, and promote sustainable use of terrestrial ecosystems, and sustainable exploration of natural resources [49];
4. A global governance sustainability focus: aims to develop and facilitate the availability of appropriate knowledge and technologies globally, as well as capacity building towards global sustainable development actions [50];
5. A global spatial planning sustainability focus: aims to promote transnational and global spatial plans to manage transnational/global natural elements (river basins, seas, mountain ranges, ice caps, forests, etc.) with a sustainable development policy approach [51].

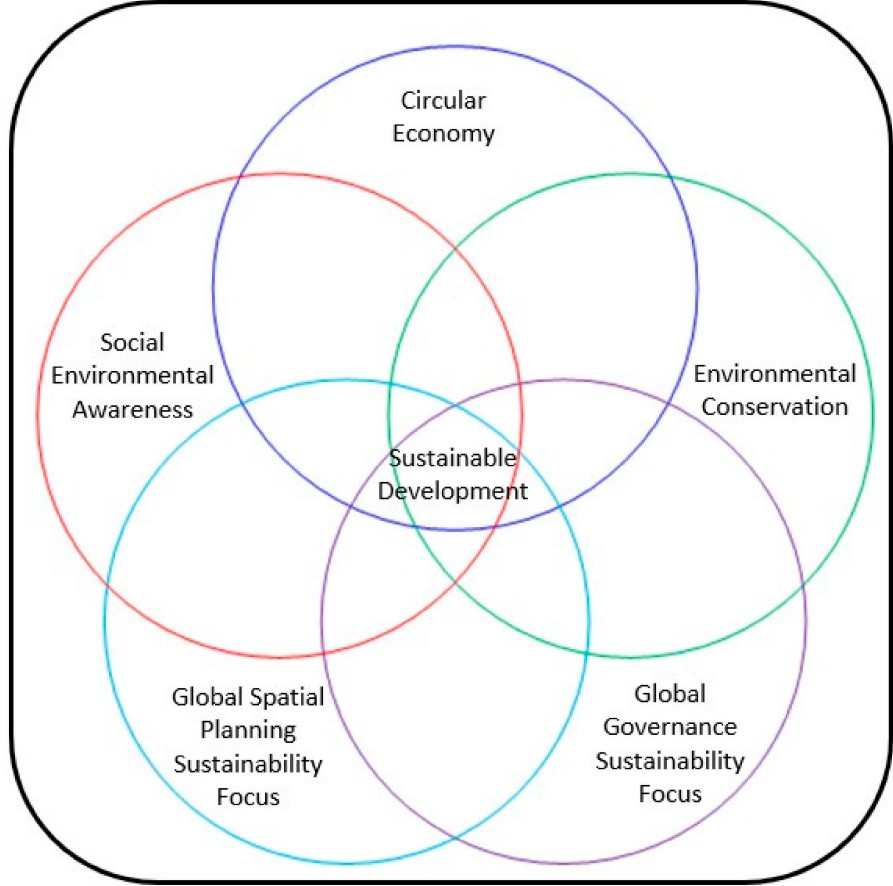

**Figure 1.** Main policy dimensions of Sustainable Territorial Development. Source: Own elaboration.

## 3. Portugal 2020: Contributions to Sustainable Development?

Since joining the EU, in 1986, Portugal has received financial support from EU funds to promote its territorial development process, in particular through the EU Cohesion Policy [5,52]. In synthesis, from the intervention's strategic priority, during the first four EU Cohesion programming periods (1989–2013), the bulk of the EU funding was allocated to (i) modernize accessibility and socioeconomic infrastructures; (ii) reinforce economic competitiveness; and (iii) support human capital and employment [53]. It is true that environmental issues have cut across these investments, especially since 1994, where the environment and urban rehabilitation were supported under Axis 3 (promoting quality of life and social cohesion). Also, in 2000, a concrete operational program (OP) was dedicated to the environment. By 2007, these environmental concerns had been included in the Territorial Valorization PO [53].

In 2014, Portugal presented the Partnership Agreement for the 2014–2020 programming phase (the aforementioned Portugal 2020). This Agreement maintains previous strategic development guidelines to supporting entrepreneurship and business innovation, strengthening research and innovation systems, increasing economic competitiveness, and qualifying human capital. Alongside these guidelines, Portugal 2020 added two specific goals to foster the modernization of the public administration and to support the shift into a low carbon and resource-efficient economy: energy efficiency and improved management of natural resources. All of these main policy goals were costed and included in four thematic objectives [54]:

- Competitiveness & Internationalization: 10.253 M€ (41%);
- Sustainable Development & Efficient Use of Natural Resources: 6.259 M€ (25%);
- Social Inclusion & Employment: 4.090 M€ (17%); and
- Human Capital: 4.327 M€ (17%).

As can be seen, the decision to allocate the largest investment share to the first goal comes from the realization that the Portuguese economy continues to struggle in the international arena, even after more than 30 years of receiving EU development and cohesion funds, as the 2008 financial crises demonstrated [55]. Indeed, only three Portuguese Nomenclature of Territorial Units for Statistics (NUTS) 2 (Lisbon Metropolitan Area—2000, Algarve and Madeira—2007) are no longer in the group of the less developed EU regions, which means most Portuguese territory is still under this undesirable EU territorial development category. A positive note, however, should be given to the 25% allocation of EU funds to the second main goal of sustainable development in Portugal 2020, and its associated aims to:

- Move towards a low carbon economy;
- Invest in renewable energy use, energy efficiency, and smart grids;
- Increase adaptability to climate change;
- Protect the coast from erosion, reduce fires and prevent flooding;
- Reduce and recycle waste and promote efficient water management.

The document which presents the Portugal 2020 strategy is very comprehensive and detailed in reporting its policy rationale to support each defined policy goal and sub-goals. In regards to the sustainable development policy goal, this document opens up a host of relevant questions related to the main constraints that Portugal still faces in this domain. These can be summarized in the: (i) high levels of energy intensity of the Portuguese economy: (ii) inefficient use and management of resources; and (iii) vulnerabilities to various technological and natural risks. As such, the proposed approach to addressing these constraints is based on three main vectors: (i) transition into a low-carbon economy, mainly associated with the production and distribution of renewable sources of energy and the promotion of energy efficiency (ii) adaptation to climate change, risk prevention, and environmental protection in following areas of intervention: water management, waste management and conservation, and enhancement of biodiversity; and (iii) the recovery of environmental liabilities and qualification, regeneration, and revitalization of the urban environment [54].

A synthetic overview of the latest Portuguese State of Environment Report [29], confirms this high energy intensity of the Portuguese economy (104.6 ton equivalent petroleum (tep)/M€ of the GDP), and a rising external energy dependency (79.7% in 2017). In this light, the allocation of EU funding to significantly increase the production of clean and renewable energy is of utmost strategic relevance. This does not signify that Portugal has not taken positive steps towards the production of renewable energy. Indeed, by 2017, 54% of the electricity produced in Portugal was generated via renewable sources of energy [56]. This was mainly due, however, to hydroelectric and eolic energy production. Indeed, the latest data show that Portugal is one of the top 20 world producers of wind energy [57]. Nevertheless, despite optimal conditions to produce solar energy [58], its production share within the overall renewable energy production is not significant (Table 1). With this in mind, the Portuguese government is backing an 800M€ plan to support the installation of 31 new solar photovoltaic energy production plants (30 in the southern and sunnier part of the country) until 2021. In total, more than a thousand MW are expected to be added to the national electric grid.

**Table 1.** Summary of energy statistics for Portugal and Europe in 2008 and 2018.

| Energy Consumption | Portugal 2008 | Portugal 2018 | Europe 2008 | Europe 2018 |
|---|---|---|---|---|
| Oil consumption in thousands of barrels per day | 293 | 236 | 16,558 | 15,276 |
| Natural gas consumption in billion cubic meters | 4.8 | 5.9 | 625.6 | 549.0 |
| Coal consumption in million tonnes of oil equivalent | 2.5 | 2.7 | 391.2 | 307.1 |
| Hydroelectricity consumption in million tonnes of oil equivalent | 1.5 | 2.8 | 134.2 | 145.3 |
| Biofuels production | 149 | 7828 | 290 | 15,949 |
| Renewable energy consumption in million tonnes of oil equivalent | 1.8 | 3.9 | 54.1 | 172.2 |
| **Energy Production (Terawatt-hours)** | **Portugal 2017** | **Portugal 2018** | **Europe 2017** | **Europe 2018** |
| Wind energy | 12.2 | 12.7 | 384.3 | 404.4 |
| Solar | 1.0 | 1.0 | 124.5 | 139.1 |
| Other renewables | 3.4 | 3.4 | 208. | 217.6 |
| Total | 16.7 | 17.1 | 717.1 | 761.1 |
| **Carbon dioxide emissions** | **Portugal 2008** | **Portugal 2018** | **Europe 2008** | **Europe 2018** |
| Million tonnes of carbon dioxide | 57.7 | 54.5 | 4939.0 | 4248.4 |

Source: Own elaboration based on Reference [57].

There have also been positive trends in Portugal regarding the reduction of carbon dioxide emissions (Table 1), representing a reduction of around 18% since 2005 levels. In this domain, the energy production and transformation sector (30%), as well as the transport sector (24%) are responsible for the bulk of these emissions [56]. In the past decade, Portugal has experienced a slight reduction of the waste production in its urban areas. However, the level of waste production has been increasing since 2014 (4.94 M tons in 2018—4.2% more than in 2017). In 2018 the waste recycling rate was around 40%, which is more or less the same as for the whole of the past decade. Nevertheless, the landfilling of biodegradable urban waste rose slightly to 46% in 2018 [56]. Finally, with regard to the environmental hazards, a striking feature which has posed problematic scenarios in several parts of the country, is the occurrence of periods of extreme drought. For instance, during the hydrologic year of 2017–2018, there were nine river basins at a hydrologic drought level in January 2018. In addition, in 2018 there were around 12,262 rural fires. Alongside this, coastal erosion has been an ongoing problem for many years. During the 1958–2010 period, the erosion of the coast line contributed to a loss of about 12 km$^2$ of territory.

Taking all the previous data together, it is possible to conclude that, from a strategic design standpoint, there is a strong policy relevance degree associated with the Portugal 2020 policy actions associated with the goals of promoting sustainable development. This is also confirmed by the revised version of the National Spatial Policy Program (PNPOT—Programa Nacional da Política de Ordenamento do Território) which provides alerts to potential negative impacts for sustainable development in Portugal from: (i) an increasing lack of water availability; (ii) increasing temperatures which will require an adaptation of the country's urban areas; (iii) an increasing rural exodus into urban areas following climatic risks; (iv) increasing vulnerability of coastal areas due to rising sea levels and saline water intrusion; (v) increasing risk of forest fires; and (vi) increasing reduction of natural habitats and biodiversity [59,60].

To tease out the wider implications of the policies' strategies, there is a need for in-depth project implementation analysis. As such, and based on the data of the Portugal 2020's executed projects until 30 September 2019, the first main overall conclusion is that the sustainable development policy goal has so far received only 14% of the Portugal 2020 program's total executed funding (Figure 2), instead of the foreseen 25%. It is true that the current programming period is still underway, and that the final execution of the available funding will probably only end by 2022. Nevertheless, the current signs are not promising for reaching the desired 25% of the EU funding towards sustainable development related projects.

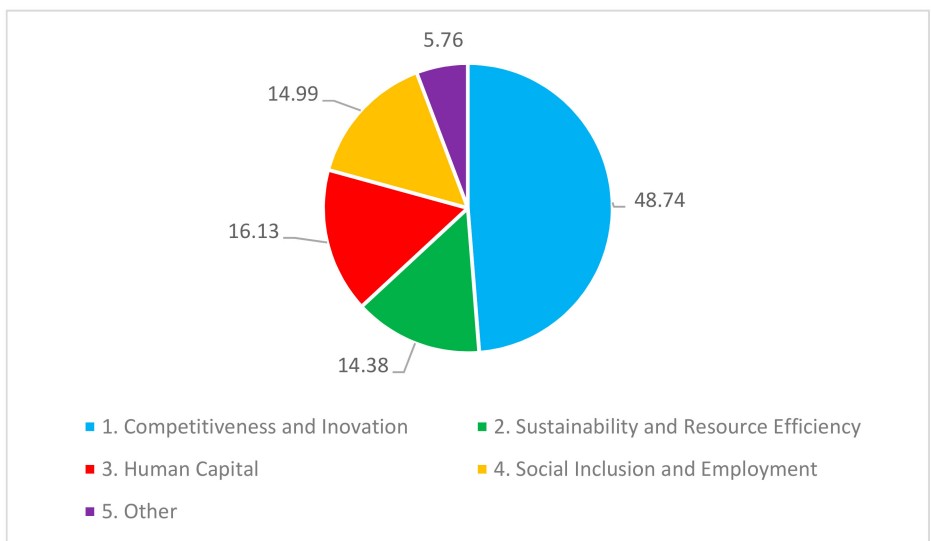

**Figure 2.** Portugal 2020 main intervention axis executed fund distribution (%) by September 2019. Source: Own elaboration based on Agency for Development and Cohesion Database until September 2019.

From a geographical standpoint, however, the distribution of the sustainable development related funding for Portugal is quite balanced, in particular across the continental territory (Figure 3). Indeed, and following a territorial cohesion [61] policy rationale, several interior and less-developed municipalities were targeted with high budget projects associated with this sustainable development intervention axis. A more detailed reading of the project database, however, presents a somewhat debatable position on the inclusion of some projects under the sustainable development flag. In this regard, a look over the flagship projects, financially speaking (Table 2), clearly demonstrates that a large share of the money associated with the SDG development goal was allocated to the modernization of transport related infrastructure. A broader analysis, however, adds the policy interventions related to water management, urban environment, and risk prevention as the main thematic operations supported by Portugal 2020 in the sustainable development arena (Table 3).

**Table 2.** Portugal 2020 sustainability development projects with more than 20M€.

| Main Policy Theme | € |
|---|---|
| Modernization of a railway-line | 139,692,149 |
| Modernization of a railway-line | 79,833,727 |
| Urban solid waste management | 79,000,538 |
| Modernization of a railway-line | 71,261,118 |
| Hydroelectric energy | 57,887,662 |
| Modernization of a railway-line | 57,857,848 |
| Modernization of a railway-line | 53,466,489 |
| Modernization of regional mobility | 52,939,764 |
| Multimodal transportation | 51,691,400 |
| Modernization of a railway-line | 51,200,521 |
| Ecologic transportation | 48,000,000 |
| Multimodal transportation | 42,634,066 |
| Multimodal transportation | 36,638,974 |
| Natural risks protection | 23,443,395 |
| Waste management | 23,360,158 |
| Ecologic transportation | 22,386,868 |
| Water treatment | 21,820,032 |
| Natural risks protection | 20,015,775 |

Source: Own elaboration based on Agency for Development and Cohesion Database until September 2019.

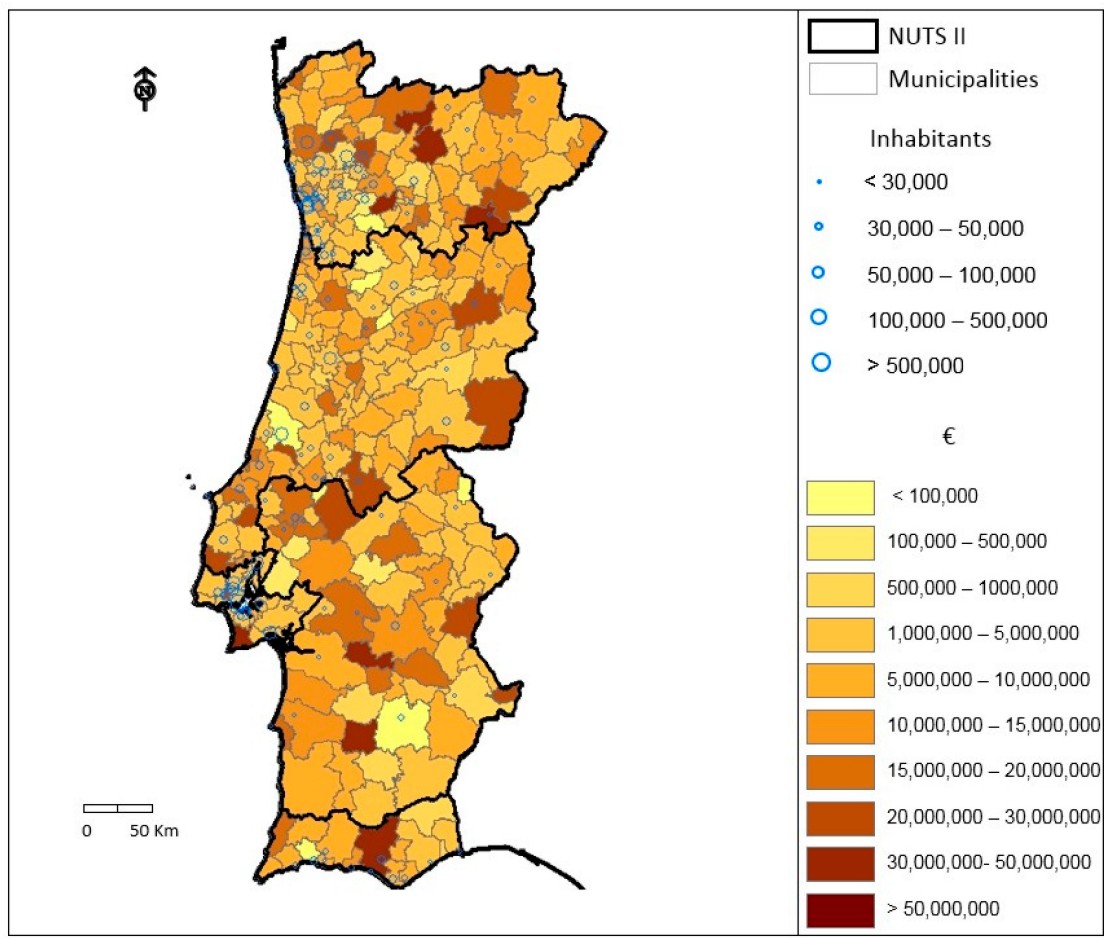

**Figure 3.** European Regional Development Fund (ERDF) distribution (€) Portugal 2020 in sustainability and resource efficiency. Source: Own elaboration based on Agency for Development and Cohesion Database until September 2019.

**Table 3.** Portugal 2020 executed funding on the Sustainability and Resource Efficiency goal.

| Policy Theme | € | % |
|---|---|---|
| 1. Multimodal Transports | 589,754,950 | 15.02 |
| 2. Climate Change and Ecosystems | 19,028,643 | 0.48 |
| 3. Renewable Energy | 93,568,459 | 2.38 |
| 4. Waste Management | 257,853,712 | 6.57 |
| 5. Risk Prevention | 472,753,724 | 12.04 |
| 6. Water Management | 658,760,186 | 16.78 |
| 7. Energy Efficiency | 290,349,074 | 7.40 |
| 8. Natural and Cultural Protection | 296,193,341 | 7.55 |
| 9. Ecologic Transports | 209,562,581 | 5.34 |
| 10. Biodiversity | 40,585,666 | 1.03 |
| 11. Cities environment | 640,240,243 | 16.31 |
| 12. Low carbon strategies | 356,967,434 | 9.09 |
| Total | 3,925,618,013 | 100 |

Source: Own elaboration based on Agency for Development and Cohesion Database until September 2019.

In synthesis, and going back to the proposed conceptual base for understanding and evaluating the policies' contribution to sustainable development, one can make the following main conclusions from the project analysis:

1.  Circular economy: There were only a few projects which specifically targeted this policy goal. In particular, they focused on increasing the selective collection of paper/cardboard, plastic/metal

and glass in municipal waste processes. The other three were closely linked with the next policy topic, which is more related to the creation of social awareness of the advantages of implementing a circular economy process. In all, this dimension has not been particularly favored by Portugal 2020.

2.  Social environmental awareness: There were several projects aligned to tackle educational and social awareness of the Portuguese population. Most were centered on alerting the population to climate change and the advantages of selective waste collection. Others raised awareness on themes like food waste, domestic and community composting, reduced hazardousness of specific packaging, forest fires, household and public administration energy savings, environmental conservation, selective collection and domestic composting, risks associated with climate change, sustainable mobility plans, and the advantages of a circular economy in urban waste. In sum, there were more than 100 projects aimed at promoting social environmental awareness of issues related to sustainable development within Portugal 2020;

3.  Environmental conservation: As seen in Table 3, natural and cultural protection received around 8% of the allocated funding for sustainable development. More fundamentally still, most of the remaining interventions have a transversal focus on environmental conservation. Hence, one can conclude that environmental conservation was a key topic for the sustainable development policy goal of Portugal 2020;

4.  Global governance sustainability: Policy-making contributes to the institutional setup, whereas territorial actors contribute to the innovative setup and informal institutional setup, hence the importance of governance aspects to policy analysis. As expected, however, this component was not particularly supported by the analyzed projects. There was, however, one project which proposed a governance and monitoring model based on outcome indicators for implementing a low-carbon development strategy. This domain is, nevertheless, slightly covered by the transnational cooperation programs supported by Portugal 2020: the Atlantic Area, South West Europe, and Mediterranean Area [51].

5.  Global spatial planning sustainability: Spatial planning is an often-mentioned domain in the analyzed projects under this sustainable development policy goal. This is especially visible in projects which aim to develop spatial planning processes related to coastal protection and the rearrangement of road and cycling traffic, parking, and pedestrian and other public spaces. Likewise, some spatial planning related interventions were focused on issues like: forest fires mitigation, urban mobility, rearrangement of road traffic, parking, water management, biodiversity, forest areas, natural heritage, and protected areas. The global spatial planning perspective is, however, clearly absent from the analyzed projects, as they focus on concrete urban/local/regional territories.

## 4. Portugal 2020 and the Production and Use of Renewable Energy Sources

Despite all the efforts from many nations in investing in the production of renewable sources of energy, recent (2019) data shows that, when it comes to the use of energy, the world still follows an unsustainable path, by moving stubbornly in the wrong direction, since "global energy demand and carbon emissions from energy use grew at their fastest rate since 2010/11, moving even further away from the accelerated transition envisaged by the Paris climate goals" [57] (p. 1). Indeed, by 2018, the renewables share of world energy production was only 4%, since almost all countries still cover their basic energy needs from fossil fuels such as oil, coal, and natural gas [62].

Demonstrably, sustainable development strategies require favoring the use of renewable energy sources as effective solutions to reduce pollution levels caused by the use of fossil fuels [62]. Here, just as with the implementation of the UN SDGs, the main responsibility to take action "remains state-centric with great room for state sovereignty, self-regulation and respect for national circumstances" [63] (p. 25). In roughly equal parts, high levels of economic growth, which is closely linked with energy consumption, tend to cause environmental degradation, thus threatening sustainable development [64].

Hence, renewable sources of energy present a concrete solution to mitigate global warming/climate change trends [30].

Portugal has an ambitious goal to use 31% of energy from renewable sources by 2020, 10% in the transport sector. By 2017, it had reached 27.3%, whereas the share of electric energy based on renewable sources reached 53.7% in 2018, against 38.5% in 2010 [53]. These numbers show a clear tendency in Portugal for an increasing use and production of renewable sources of energy, in overall terms, as the data from Table 4 shows. However, the steady increase of eolic-based energy contrasts with the irregular production of hydroelectric-based energy, since Portugal has a quite variable weather system, prone to dramatic yearly changes in precipitation values. Moreover, as previously mentioned, it is also clear that Portugal has not been capable of harnessing its tremendous photovoltaic energy production potential [58], unlike other south European countries [65].

**Table 4.** Annual production of renewable sources of energy (GWh) in Portugal (2010–2019).

|  | 2010 | 2011 | 2012 | 2013 | 2014 | 2015 | 2016 | 2017 | 2018 | 2019 |
|---|---|---|---|---|---|---|---|---|---|---|
| Hydroelectric | 16,547 | 12,114 | 6660 | 14,868 | 16,412 | 9800 | 16,916 | 7632 | 13,628 | 8814 |
| Eolic | 9182 | 9162 | 10,260 | 12,015 | 12,111 | 11,608 | 12,474 | 12,248 | 12,617 | 12,894 |
| Biomass | 2226 | 2467 | 2496 | 2516 | 2578 | 2518 | 2481 | 2573 | 2558 | 2624 |
| Biogas | 100 | 161 | 210 | 250 | 278 | 294 | 285 | 287 | 271 | 245 |
| Urban solid waste | 577 | 592 | 490 | 571 | 481 | 584 | 610 | 632 | 573 | 587 |
| Geothermic | 197 | 210 | 146 | 197 | 205 | 204 | 172 | 217 | 230 | 206 |
| Photovoltaic | 215 | 282 | 393 | 479 | 627 | 799 | 871 | 993 | 1006 | 1248 |
| Total | 28,754 | 24,692 | 20,411 | 30,610 | 32,453 | 25,514 | 33,503 | 24,309 | 30,637 | 26,366 |

Source: Own elaboration based on Reference [66]. Note: 2019 until September.

Based on the current panorama of renewable energy production, in a country like Portugal, which imports all its oil, gas and coal (around 11.5% of the total imports in 2019 [67]) for energy production, and which has an untapped potential to further explore renewable sources of energy (solar—mostly in the south and interior part of the country and in particular in urban areas, offshore eolic and tidal, and biomass—near forest areas), one would expect that a large share of the Portugal 2020 funds would have been allocated to develop the production and use of renewable sources of energy. Strangely, this is not the case. Indeed, the analysis of the Portugal 2020 project database (Table 5) allows for the following main conclusions:

1.  Funding: the share of the allocation of funds for the production of renewable sources of energy is particularly low in view of the country's potential in this domain. Here, hydroelectric is, by far, the most financed source of renewable energy, in a project located on the island of Madeira (Calheta).

2.  Support for solar energy: surprisingly, not a single project was dedicated to exploring the country's potential to be one of the world's leaders (as it is with the production of wind energy) in exploring solar sources of energy (photovoltaic, thermal), namely in the southern part of the country and in urban areas, such as: brightfields [68]; on buildings via rooftop PV cells and water-heating systems [69]; transport [70]; roads, sidewalks, vacant land at industrial sites; large rooftop areas of car parks and shopping centers; and on degraded or contaminated land [71].

3.  Off-shore energy production: there is an interesting project, which is part of Portugal 2020, to assess the potential impacts of the implementation of off-shore sources of energy (wind and wave) in a country with a vast oceanic coast. In our opinion, this is largely insufficient in light of the country's potential to explore these energy sources. In this regard, Portugal could follow the examples of other countries like the United Kingdom [72], even though the Portuguese continental shelf is not as vast and shallow as the shelf of the North Sea [73];

4.  Biomass and others: There are three projects aiming at exploring the potential of Portuguese biomass, which is still significant in view of the Portuguese forest area, and despite the associated annual forest fires [74]. Another positive note, in our understanding, is given to the exploration of hydrogen as a potential source of clean energy, namely in vehicles.

**Table 5.** Portugal 2020 executed funding on the promotion of production and distribution of renewable sources of energy sub-goal.

| Project Main Goal | Source of Energy | € |
|---|---|---|
| Identify optimal sources of biomass | Biomass | 227,884 |
| Optimization of biomass use | Biomass | 47,482 |
| Assess the potential and impact of hydrogen use | Hydrogen | 58,881 |
| Assess the potential and impact of hydrogen use | Hydrogen | 142,648 |
| Storage of compress air | - | 130,420 |
| Assess the impacts of the use of offshore energy | Wind and tides | 355,288 |
| Expansion of hydroelectric plant | Hydroelectric | 57,887,662 |
| Construction of a biomass plant | Biomass | 8,385,091 |
| Construction of a battery storage center | - | 11,500,000 |
| Construction of a battery storage center | - | 973,850 |
| Total | | 79,709,206 |

Source: Agency for Development and Cohesion Database until September 2019—Own elaboration.

In sum, the Portugal 2020 contribution to improving the use and production of renewable sources of energy is limited and insufficient, taking into account the untapped potential of the Portuguese territory in this domain. Additionally, and this is extensive to the use of EU funding in Portugal, the policy intervention logic is one of fuelling pinpoint project proposals instead of supporting a clear development strategy which boosts the territorial development potential of the country. In this regard, and in our opinion, Portugal 2020 looks to be another lost opportunity to place Portugal in the group of the most developed European countries, in particular, by smartly exploring its main territorial sustainable development potential, especially in the renewable energy policy cluster. To achieve this, there is a need to ring-fence EU funding to key territorial sustainability development areas, such as the promotion of urban sustainability development strategies [75] and off-shore wind and tidal related energy production.

## 5. Conclusions

This paper has shone some light on the expected contribution of Portugal 2020 to promoting sustainable territorial development processes. The analysis advanced a novel proposed theoretical approach which considers the sustainable territorial development as a five-dimensional concept, which should contribute to supporting a circular economy, social environmental awareness, environmental conservation, global governance sustainability, and global spatial planning sustainability, applied to the Portuguese case.

Put simply, and based on a detailed project analysis of the executed Portugal 2020 projects until September 2019, it was possible to conclude that the initial goal to allocate around 25% of the total funds from Portugal 2020 to promoting sustainable development is a far cry from the current execution (14%). Worse still is the lack of strategic vision for the sustainable territorial development of Portugal, in view of its territorial needs and potentials in the medium and long term. This is clear when analyzing the lack of support for the exploration of solar energy, namely in urban areas, as well as the limited support to promote the implementation of a circular economy. Indeed, from analyzing the Portugal 2020 project database it is possible to verify a clear continuation path from past EU Cohesion Policy programming periods, in which the project selection does not follow a clear overall strategic development vision for Portugal. Rather, the approved projects follow a rationale for pin-pointing and solving local/regional public/private development interests and visions, whose relevance and significance can be questioned for the overall and sustainable development of Portugal.

On a positive note, however, Portugal 2020 has supported several projects aiming at building up a social environmental awareness, in particular by alerting the Portuguese population to the consequences associated with global warming (the predicted rise of the ocean is expected to affect many populated areas in the Portuguese territory in the next decades ) and the need to increase waste

recycling practices. Likewise, the measures taken to promote environmental conservation were applied with sufficient strength, relevance, and in a balanced way across the Portuguese territory.

One result that can animate a controversial debate is the support for the modernization of the national railway system, which absorbed 15% of Portugal 2020's sustainability and resource efficient goal funding. this is because the supporters of these investments can highlight the excessive investment in road and highway building in the previous EU Cohesion Policy programming periods in Portugal, vis-à-vis the comparable reduced investment in modernizing the railway system, which is viewed as a more green and sustainable form of transport accessibility infrastructure. On the other hand, the bulk of the investments in the railway system were not in building new railway connections, which add additional alternatives for existing road connections. Moreover, the replacement of old railway rolling stock by more modern and energy efficient stock has not been fundamentally supported.

It is true that the amount of available money to support public policies is crucial to help achieving the desired policy goals. Nevertheless, as the history of EU Cohesion Policy has shown time and again, its positive impacts depend on multiple factors that go beyond the monetary one. Crucially, these impacts are also dependent on how and where (development dimensions/components and location) the money is allocated. In the case of Portugal 2020, there is no doubt that it has contributed to promoting sustainable territorial development, in particular in the environmental protection and the social environmental awareness dimensions. However, the estimated financial misdistribution for this 'sustainable development' pillar of Portugal 2020 has not reached the intended funding (14% vs 25% when almost 50% was allocated to promoting the economic competitiveness policy goal), nor does it look to have been implemented with a clear strategic vision which addresses central components of sustainable development in Portugal, such as the support for a robust circular economy strategy and urban sustainability. Similar conclusions could be drawn in other EU Member States as well, if the applied methodological rationale is used there. In particular, countries that are structurally similar to Portugal can profit from this analysis in order to increase the effectiveness of their sustainable territorial development policies. This can be done by following a more targeted strategic policy intervention aiming to foster their territorial capital, instead of supporting local/regional and private interests disconnected from an overall territorial sustainable development strategy.

As usually happens, there is a need to wait for the conclusion of Portugal 2020 to produce a final and conclusive study on its contribution to sustainable territorial development in Portugal. Alongside the use of a project and literature analysis, interviews with major (national, regional, and local) players on the studied domain could help to add more precision to this paper's conclusions. Even so, there is clearly more than enough data to support its main conclusions that there is a need for a more targeted and strategic project selection rationale in the future EU Cohesion Policy period in Portugal, in order to increase the efficiency of the allocated funds in the mentioned crucial domains of the sustainable territorial development.

**Author Contributions:** The author is the lead for all aspects of this research.

**Funding:** This research received no external funding.

**Conflicts of Interest:** The author declares no conflict of interest.

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
