# Peer review of "Portugal 2020: An Effective Policy Platform to Promote Sustainable Territorial Development?"

_sustainability, doi:10.3390/su12031126_

Round 1

Reviewer 1 Report

Sustainability-707376:

The author is interested in sustainable territorial development, which fits well within the scope of Sustainability. In this context, the author aims to analyze an application of EU cohesion policies to maintain sustainable territorial development in Portugal. This provides an interesting and state-of-the-art evidence for sustainable territorial development which would help policy-makers advance their strategies.

Nevertheless, I believe that the manuscript has a greater potential to be developed, especially regarding the value added. Let me start with a constructive critique:

In the introductory part, the author claims that “Sustainable development has been the most important concept developed by mankind during the twentieth century and early twenty-first century”. This may be the personal opinon of the author, but I would not start a scientific article with such sharp and subjective claims - even though sustainability is very important, evidence for it can be found in earlier texts published centuries before the 20th century. For your introductory part, you would need a deeper motivation to justify your case study. You should do this before you mention the EU cohesion policy.
As it is known, EU cohesion policy aims to reduce disparities in the level of development between the regions. This is done in terms of strengthening economic and social cohesion. This policy instrument is formulated for 28 nations, aiming to achieve similar results in structurally similar regions. This should be your starting point. This approach goes back to evolutionary economics, and states that “two economic processes are structurally similar, if their properties constituting the definition of ‘structure’ are similar”. For achieving economic and social cohesion in structurally similar regions, we need case studies, empirical evidence, policy implications and so on… Viewing this as your point of departure, you can “justify” why you are conducting your research and why you are focusing on Portugal. I would suggest to take a look at https://doi.org/10.3390/su11051437 and https://doi.org/10.3390/su11061771 for incorporating this approach in your introductory part and your literature review. In the literature review, the author needs a clearly framed research question. I believe that structural similarities approach may help the author to do it, since the author would ask how we can achieve sustainable territorial development in structurally similar territories – instead of telling that the focus is on Portugal and there is no value added for any other country. This can help the author to have a more focused literature review. I believe that the starting point should be the approach by https://doi.org/10.1007/978-3-319-96032-6_5 for the fact that sustainable territorial development is always an interplay between institutions and innovation. In the case of the author, the formulated policy goals also reflect this interplay. Policy-making contributes to the institutional setup, territorial actors contribute to the innovative setup and informal institutional setup. In addition, I would expect from the author to formulate a research gap and a research question at the end of the literature review. I believe that the main part of the analysis, that is chapter 3, is conducted quite well and can be enriched by the approaches proposed above. Nevertheless, I must also point out that I neither see any explanation for nor any need for figure 3. It appears like out of nowhere and without any comments, this only confuses the readers. Finally, I would like to ask what we learn from this approach when we are not particularly focusing on Portugal. In other words, what is the transfer value added of the manuscript? The author does not say anything about this, but I believe that those countries which are structurally similar to Portugal can profit from this. Furthermore, I would also add some limitations (focus on a single case) and some future research aspects to this research. A minor comment: Subtitles- such as those in chapter 3, should be represented in the MPDI format. The same applies for tables and figures.

Good luck!

Author Response

The author thanks the reviewers and the editor for the constructive remarks which he used to revised the text. In all, their remarks were followed closely. In detail:

- The introduction was largely reshaped by including additional text following the reviewer’s suggestions as well as the structure with the elimination of sub-titles

- The figures, tables and references are following the journal rules

- Figure 3 was placed in the second section

- All references were fully revised and a few added

-  Tables were changed in their configurations

- The references acronyms are expressed on the reference text

- The references in the tables follow the journal rules

-  The conclusions were completed with the mention of the potential gains from the proposed methodological approach to other countries.

- Some parts of the text were rearranged following the editor remarks on the originality report (Lines 79-85 & 120-123 & 213-221)

Reviewer 2 Report

This is a challenging article on a topic of high interest and it is my pleasure to review it.

The paper raises interesting topics and stimulates beneficial debates on the topic of Sustainable Territorial Development, making it relevant for practitioners, policy-makers and scholars. We consider that this paper is likely to become a benchmark in Sustainable Territorial Development through EU funds, especially useful for the new EU member states with emerging economies, that can face similar challenges in the next period of time.

Methodology and approaches are interesting, systematic and comprehensive.

I would have some considerations and suggestions for improving the quality of the article. This does not diminish the merits of the paper, but there are recommendations which might be worth taking into account.

The paper’s structure is not very clear – i.e. on chapters and subchapter. There are several classifications, titles and subtitles, underlined or with different letter size, that, sometimes affect reading fluency.

It is not very clear why (on page 10) the author returns to the proposed conceptual base, explaining in detail the proposed theoretical model, although, in the previous pages, he developed quantitative analyses on the Portugal’s achievement. It was probably more understandable if these theoretical explanations forego the analysis, and if the case, the model was tested and discussed, on real data, in results and partial conclusions.

It is recommended that the sources of certain tables to be explicitly written, not as abbreviations or figures indicating cited works, for example: Table 1. Source: (BP, 2019) or Table 4 Source: [73] - Own elaboration.

Abbreviations need to be explained, not to be supposed (for example, by searching in final References) ex DGEG, BP.

In Table 1 POR and EUR in table head - is it about Portugal and Europe? It is recommended to write their full name.

The final references, in some cases, are incompletely drafted, with abbreviations etc., and not in the mdpi style.

Thank you for the opportunity to review this article and good luck!

Author Response

(The authors gave the same response as above.)

Round 2

Reviewer 1 Report

This is a good and interesting paper. However, I was hoping for more precision around the use of the term "structural similarities", as per my previous comments.

Regarding your observations in lines 480 thru 483, you need to explain what you mean. What should other countries which are structurally similar to Portugal learn from your analysis? This is a question you still leave vague in your concluding remarks.

A minor comment: I believe that reference [7] is wrong, since structural similarities approach is not due to Gagliardi and Percoco as the author claims, but due to Lehmann-Waffenschmidt and Erkut, see DOI: 10.18559/SOEP.2016.5.6 , https://doi.org/10.3390/su11051437 and https://www.osti.gov/etdeweb/biblio/21097574 for the correct references.

Author Response

I would like to thank the reviewer help and constructive comments.

- A few remarks were added to the text to address the reviewer comments

- Reference replaced.